# Bilateral Abducens Nerve Palsy Due to Extensive Cerebral Venous Sinus Thrombosis

**DOI:** 10.3390/medicina55040115

**Published:** 2019-04-23

**Authors:** Krunoslav Buljan, Gordan Šarić, Dennis Czersky Hafner, Romana Perković

**Affiliations:** 1Department of Neurology, Osijek University Hospital, Faculty of Medicine, University of Osijek, 31000 Osijek, Croatia; dennisczersky.hafner@gmail.com (D.C.H.); romana.per@gmail.com (R.P.); 2Department of Radiology, Osijek University Hospital, Faculty of Medicine, University of Osijek, 31000 Osijek, Croatia; saric.gordan@gmail.com

**Keywords:** abducens palsy, cerebral venous sinus thrombosis, protein S deficiency, papilloedema

## Abstract

Cerebral venous sinus thrombosis (CVST) is a relatively rare condition. We present a case of an acute aseptic thrombosis of the sagittal, transverse and sigmoid sinus in a puerperium patient with protein S deficiency. The specifics of the case include high intracranial pressure (ICP) caused by sinus thrombosis with typical symptomatology and bilateral papilloedema, which also manifested in transient bilateral abducens nerve palsy and, consequently, bilateral horizontal diplopia. The recovery of the cranial nerve function occurred 3 to 4 weeks after it was initially reported. Prompt and adequate anticoagulant therapy contributed to the almost complete recanalization of the dural venous sinus thrombosis and a positive outcome of the disease.

## 1. Introduction

Cerebral venous sinus thrombosis (CVST) is a relatively rare life-threatening condition with variable symptomatology, similar to numerous other neurological disorders. Worldwide estimated annual incidence is 3–4 cases/1 million people [1]. Approximately three-quarters of adults with CVST are women [1]. CVST risk factors include severe dehydration, final trimester of pregnancy, puerpera, genetic disorders of the clotting cascade [2], connective tissue disease, malignancy, contraceptive use and infections such as sinusitis, otitis, and mastoiditis [3]. Protein S is a vitamin K-dependent protein that prevents clotting by acting as a cofactor for activated protein C [4]. Inherited or acquired protein S deficiency leads to the prothrombotic state, with plausible thrombosis [4]. Papilloedema refers to optic disc oedema associated with increased intracranial pressure (ICP) [5]. The sixth cranial nerve is the most vulnerable to increased ICP due to its extended intracranial course, and due to its sensitive position against the sphenoid ridge and the petrous portion of the temporal bone [6]. We describe the case of an extensive CVST in a young puerperium woman, with an acute clinical onset consistent with focal somatosensory seizures, with symptomatology of increased ICP, bilateral papilloedema, and the rare clinical occurrence of bilateral sixth nerve palsy as a consequence of increased ICP.

## 2. Case Presentation

A 23-year-old woman was urgently hospitalized due to the acute onset of headaches, nausea, and vomiting lasting over 5 days, with left-side extremity paraesthesia. Two weeks prior to the hospitalisation, the patient underwent a C-section in her 31st week due to foetal hypoxia. The baby was safely delivered. Prior to that, the patient was monitored by ultrasound due to the absence of detectable vascularization in several limited regions within the placental tissue, with the diameters of 35, 19 and 11 mm. She was administered antibiotics (amoxicillin + clavulanic acid) postpartum due to a mild temperature, but her temperature was not high enough to indicate infection. Two days prior to hospitalisation she was urgently screened neurologically because of a 3-day long headache. At that moment, she was without neurological deficits. The occipital headache was transiently lower in intensity after she received analgesic therapy. On the day of the admission, the patient had an intermittent short-term numbness in the left hand, left leg, and she reported she once felt numbness on the left side of her face.

Neurological examination indicated mild palsy of the left hand. A confrontation visual field test detected inferior homonymous quadrantanopia. Her gait was normal, not paretic. The patient was afebrile. Meningeal signs were negative. Blood pressure was 120/85 mmHg. Laboratory values upon admission are presented in Table 1.

During the last period of the pregnancy, reduced levels of free protein S (fPS) antigen (0.43) (1), and protein C path (or protein C global or activated protein C resistance) (0.77 normalized ratios (reference range >0.80)) were detected. There was no mutation in the genetic markers for thrombophilia (Factor V Leiden, Prothrombin *MTHFR* and *PAI-1*). Pathohistological tests of the placenta conducted after birth indicated placental infarction.

The acute thrombosis of the right transverse sinus, the right sigmoid sinus, and the sagittal sinus (Figure 1A) were detected by urgent computed tomography (CT) of the head. The CT did not detect brain oedema or any signs of focal lesion. Immediate magnetic resonance imaging (MRI) of the brain was performed and the results were normal. Magnetic resonance venography (MRV) confirmed the absence of flow in the right transverse and right sigmoid and sagittal sinuses as a consequence of thrombosis (Figure 1B).

Immediately following neuroimaging detection of a CVST, low-molecular-weight heparin therapy was administered (dalteparin subcutaneously twice daily at 7500 IU), followed by analgesics and folic acid substitution therapy. As recommended by the infectious diseases specialist, antibiotic therapy (ceftriaxone and vancomycin) was administered for a total of 12 days. Fundoscopic examination detected initial bilateral optic disc oedema, without visible peripapillary or retinal bleeding. The otorhinolaryngologic examination excluded any infectious diseases in the respective area. From the second day of the hospitalisation onward, the patient did not report paraesthesia or sight deficit. Electroencephalogram during the interictal period detected generalized cerebral dysrhythmia without lateralization. The symptomatology of increased ICP and papilloedema continued, which lead to introducing acetazolamide on the eighth day of the hospitalisation at an overall daily dose of 500 mg, and for a few days, mannitol 10% was given intravenously. Eight days after the initial examination, fundoscopy findings and optical coherence tomography (OCT) indicated worsening of bilateral papilloedema. Voluminous veins and flame-shaped haemorrhages were noticed at the edges of the right optic nerve papilla as well as one larger flame-shaped haemorrhage on the left temporal side.

At the beginning of the second week of the hospitalisation, the patient reported horizontal diplopia when looking left, with visible left lateral rectus muscle insufficiency (Figure 2A). Five days later, we noticed a deficit in the function of the right lateral rectus muscle (Figure 2B). At that point, the patient reported double vision in both horizontal directions. The Hess–Lancaster test confirmed weakness of both lateral rectus muscles (Figure 2C). Visual field examination performed according to Goldmann demonstrated normal findings.

In the third week, there was a significant reduction and then a complete cessation of headaches. A follow-up MRV was performed 18 days after the initial one, and revealed partial recanalization of the dural venous sinus thrombosis (Figure 3A,B).

In the fourth week of treatment, the recovery of the function of the left lateral rectus muscle was noticed at the same time as the intensity of the left-sided horizontal diplopia receded and ceased. Ten days later, the recovery of the right rectus muscle function was noticed and the horizontal diplopia completely ceased. The results of the follow-up Hess–Lancaster test were normal. Outpatient anticoagulant therapy was continued with warfarin.

Two months after the beginning of the treatment, fundoscopy results and OCT recorded complete withdrawal of the papilloedema. At the follow-up neurological examination three months after the onset of the disease, the patient was without neurological symptoms and without a bulb motility disorder. The follow-up MRV, three and a half months after the initial one, indicated further significant regression of the CVST. Along with the clear findings of the sagittal sinus, there was minimal marginal contour irregularity in the right transverse sinus and minor residual strip defects in the right sigmoid sinus (Figure 4). Six months after CVST, warfarin therapy continued and the fPS antigen level was persistently low at 0.36 (1).

The study protocol was approved by the institutional review board, with the ethical code number R2-1281/2019.

## 3. Discussion

So far, the literature does not describe a case of bilateral abducens palsy in aseptic CVST in puerperium patients with protein S deficiency. Research shows that bilateral abduction palsy is quite rare compared to its occurrence unilaterally. Balasubramanian et al. and Lang et al. described CVST cases with bilateral abducens palsy as a complication in chronic suppurative otitis and mastoiditis [7,8]. In those cases, unlike in ours, the bilateral sixth nerve palsy was one of the initial CVST symptoms. Other causes of bilateral abducens palsy described in the literature, such as diabetes mellitus, idiopathic intracranial hypertension, Duane syndrome, trauma [9] and subarachnoid haemorrhage [10], were not present in our patient. Inferior petrosal sinus thrombosis is described in the literature as an extremely rare possible cause of compression of the ipsilateral sixth nerve [11] but not as a cause of a bilateral lesion. MRV in our patient did not clearly detect inferior petrosal sinus thrombosis. Therefore, in our patient, bilateral abducens palsy can only be explained as a result of increased ICP and consequential compressive damage. This is corroborated by the period of simultaneous increased ICP, most pronounced papilloedema and clinical representation of bilateral abducens palsy. In our case, the headache, as the most significant symptom of the increased ICP, preceded bilateral papilloedema and sixth cranial nerve palsy, as well as signs of ICP, but it also receded prior to the other pathologic signs described.

Transient numbness in our patient was consistent with focal somatosensory seizures. The opinion of O’Rourke et al. that thrombosis of the upper sagittal sinus is mainly aseptic [5], is corroborated by this case report, in which the patient also suffered an aseptic thrombosis (absence of clinical and radiological signs of otitis, sinusitis or mastoiditis, and a normal serum procalcitonin level at the onset of the disease).

CVST in protein S deficiency, in comparison to other thrombotic incidents, has been described in only a few case reports [4]. Makanjuola et al. described the case of an extensive CVST with protein S deficiency in a pregnant woman [12]. Narayan et al. in people with CVST of both sexes, detected protein S deficiency in 12% of the respondents [13]. In our patient, the prothrombotic state associated with protein S deficiency clinically manifested in the third trimester by placenta infarction and again in the puerperal period by CVST. The onset of CVST symptoms in our patient occurred five days prior to the diagnosis, which corresponds to the results of the research by Kashkoush et al. conducted on 66 subjects, who were either pregnant or in the puerperal period. According to Kashkoush et al. the largest number of CVST cases was diagnosed 4.2 to 7.6 days after the initial onset of the symptoms [14].

Elevated D-dimer results, which may indicate thrombosis [5], gradually normalized in our case, along with the process of regeneration of sinus thrombosis. Similarly to our patient, Verma et al. described early recanalization of venous sinuses thrombosis with protein S deficiency after 15 days of anticoagulant therapy [4]. According to data available in the literature, a high recanalization grade of CVST independently predicts positive neurological outcome [15], while lack of venous recanalization is associated with worse clinical outcome [16]. A study on 50 subjects indicated that patients with multiple sinus thromboses have a higher case fatality rate [17]. Unlike the results of Patil et al. [17], in our patient with bilateral dural sinus thromboses and bilateral papilloedema, the clinical outcome of the CVST was excellent (modified Rankin score 0).

## 4. Conclusions

In a case of a relatively acute onset of intense postpartum headaches and stroke-like symptoms, it is reasonable to suspect CVST. Bilateral papilloedema and bilateral abducens palsy could be clinical signs of increased intracranial pressure in an extensive CVST. Timely and effective anticoagulant therapy in CVST significantly contributes to venous recanalization and a good clinical outcome of the disease.

## Figures and Tables

**Figure 1 medicina-55-00115-f001:**
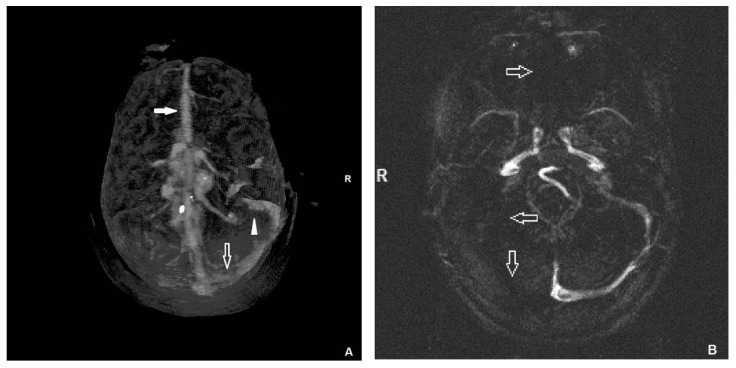
(**A**) Computed tomography of the head shows acute thrombosis of the right transverse sinus (arrow), the right sigmoid sinus (arrowhead) and the sagittal sinus (filled arrow). (**B**) Magnetic resonance venography confirmed the absence of flow in the right transverse, right sigmoid and sagittal sinuses (arrows) as a consequence of thrombosis. R = right.

**Figure 2 medicina-55-00115-f002:**
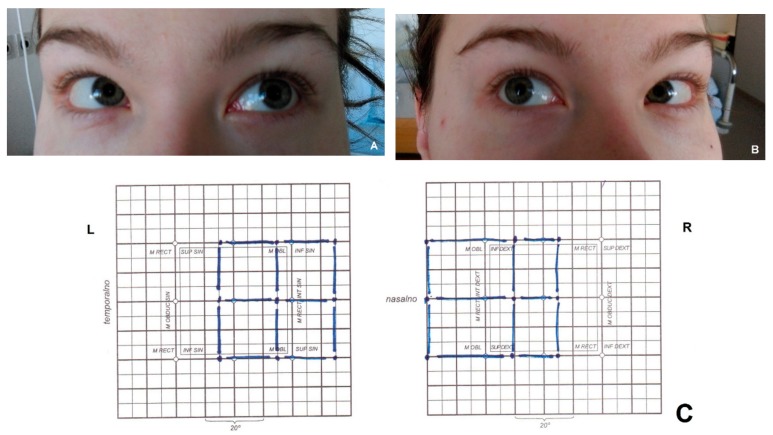
Patient with bilateral abducens palsy due to extensive cerebral venous sinus thrombosis. Images show inability of (**A**) the left eye to turn out in left gaze and (**B**) the right eye to turn out in right gaze. (**C**) Bilateral abducens palsy was confirmed by the Hess–Lancaster test.

**Figure 3 medicina-55-00115-f003:**
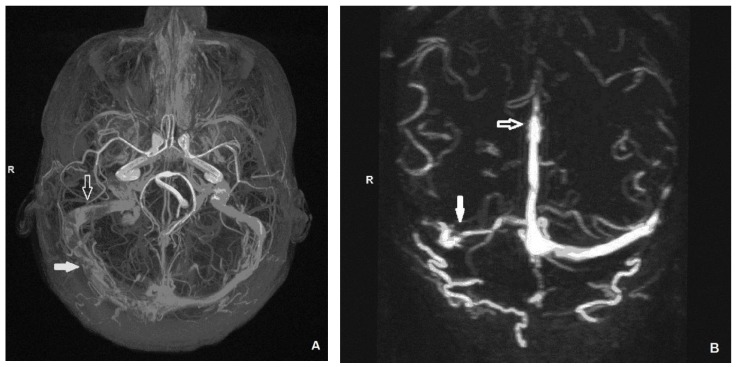
Follow-up magnetic resonance venography (18 days after the initial one) revealed partial recanalization of cerebral venous sinuses thrombosis; (**A**) sigmoid (arrow) and transverse (filled arrow); (**B**) sagittal (arrow) and transverse (filled arrow).

**Figure 4 medicina-55-00115-f004:**
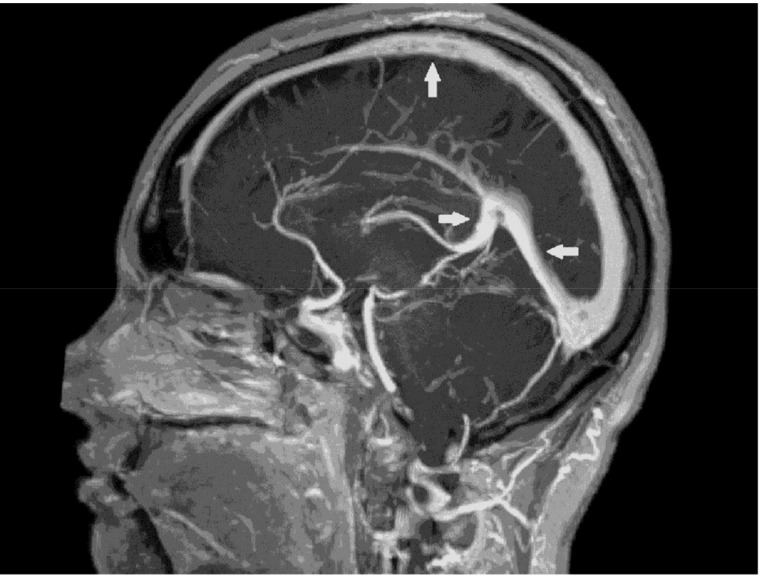
Follow-up magnetic resonance venography (3.5 months after the initial one) indicated further significant regression of the cerebral venous sinus thrombosis (arrows).

**Table 1 medicina-55-00115-t001:** Laboratory values upon admission.

Clinical Laboratory Test	Value	Reference Range
D-dimer, µg/L	2454	0–500
Heparin anti-Xa activity, IU/mL	0.71	0.85–1.37
Factor VIII (1)	2.31	0.50–1.49
Free protein S antigen (1)	0.55	0.60–1.41
Prothrombin time, s	1.15	0.70–1.27
Thrombin time, s	18.7	14.0–21.0
Fibrinogen, g/L	2.7	1.8–3.5
Activated partial thromboplastine time (1)	0.84	0.80–1.20
Antithrombin activity (1)	1.21	0.85–1.37
Thrombocytes, 10^9^/L	601	158–424
Procalcitonin, ng/mL	0.02	0.00–0.50
Homocystein, µmol/L	9.2	5.0–15.0
Folic acid, nmol/L	6.2	7.0–46.4
Total cholesterol, mmol/L	7.34	<5.00
Low-density lipoprotein cholesterol, mmol/L	4.97	<3.00

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
