# Peer review of "Bilateral Abducens Nerve Palsy Due to Extensive Cerebral Venous Sinus Thrombosis"

_medicina, 2019, doi:10.3390/medicina55040115_

Round 1
Reviewer 1 Report
Interesting case and findings! A few suggestions/questions:
Page 2 Line 57 – how was the field defect evaluated? Confrontation, Humphrey, Goldmann? Was the visual field retested at a later date to confirm stability/improvement?
Page 2 Line 70 – these labs may be better represented in a table than listed in the text.
Page 3 Line 104 – were these measurements measured with prism diopters?
Page 5 Line 144 – Symonds syndrome and Wildervanck syndrome and not common names of these conditions. I would just call them idiopathic intracranial hypertension and Duane syndrome, respectively, and drop the other terms.
General question:
(1) Was consideration given the lumbar puncture?
(2) Did she at any time of transient visual obscurations or pulsatile tinnitus?’
(3) Do you have photos of the papilledema? OCT images?
Author Response
REVIEWER 1 - ANSWERS
English language and style
( ) Extensive editing of English language and style required
( ) Moderate English changes required
(x) English language and style are fine/minor spell check required
( ) I don't feel qualified to judge about the English language and style
Comments and Suggestions for Authors
Interesting case and findings! A few suggestions/questions:
Page 2 Line 57 – how was the field defect evaluated? Confrontation, Humphrey, Goldmann? Was the visual field retested at a later date to confirm stability/improvement?
The visual field defect is detected by the confrontation method as part of the neurological examination. Normalization of the visual field on the second day of hospitalization was detected in the same way. During the hospitalization, a review of the vision field by Goldman was performed, which did not show any visual field deficits.
• „Confrontation visual field test“ – added to the text because of the explanation of the test with which the visual field deficiency is detected.
• The sentence added to the reviewer's request: „Visual field examination performed according to Goldmann demonstrated normal finding“.
Page 2 Line 70 – these labs may be better represented in a table than listed in the text.
According suggestion, laboratory findings at admission are represented in a table.
Page 3 Line 104 – were these measurements measured with prism diopters?
No. Prism diopters in our work conditions are not routinely monitored in all patients with paralytic strabismus. Testing with prisms are usually done when no recovery occurs and an operating procedure is required.
Page 5 Line 144 – Symonds syndrome and Wildervanck syndrome and not common names of these conditions. I would just call them idiopathic intracranial hypertension and Duane syndrome, respectively, and drop the other terms.
According suggestion „Symonds syndrome and Wildervanck syndrome“ is left out.
General question:
• Was consideration given the lumbar puncture?
Lumbar puncture, or analysis of the liquor, was not done because there was no clinical suspicion of intracranial infection or intracranial hemorrhage. On the other hand, lumbar puncture in a patient with high intracranial pressure can be potentially dangerous.
• Did she at any time of transient visual obscurations or pulsatile tinnitus?’
According to anamnestic data, neither before nor during the course of the disease such disturbances were not recorded.
Do you have photos of the papilledema? OCT images?
Yes, we have the patient's OCT images. Ophthalmological description of the images is cited in the manuscript. If the reviewer and the editorial are of the opinion that the OCT images should be added in the manuscipt, it can be done.
Reviewer 2 Report
This is an interesting case report which should be focused on its novel aspect: the bilateral VI nerve palsy
Both the introduction and discussion should be modified to concentrate on this aspect.
Protein S deficiency and puerperium are well known risk factors for CVT, so there is nothing new about puting emphasis on those aspects
Because of that I strongly suggest to modify the tittle and remove "in a pueperium patient with protein S deficiency"
Abstract - line 2 "have presented" should be "present"; lines 8-9 remove the sentence on the prognostic indicators because the statement is not correct. . Multiple venous occlusions and papilloedema are not prognostic indicators for death or disability (see CVT risk score, also available as a free App). Papilloedema is only a por prognostic sign for visual impairment.
Case report - delete all not essential negative or normal findings
Transient numbess were probably focal seizures. CVT do not cause TIAs. Was na EEG performed?
Summarize the follow up
Discussion
remove the mention to TIAs and to the poor prognostic signs
Author Response
REVIEWER 2 - ANSWERS
Open Review
(x) I would not like to sign my review report
( ) I would like to sign my review report
English language and style
(x) Extensive editing of English language and style required
( ) Moderate English changes required
( ) English language and style are fine/minor spell check required
( ) I don't feel qualified to judge about the English language and style
Yes | Can be improved | Must be improved | Not applicable | ||
Does the introduction provide sufficient background and include all relevant references? | ( ) | (x) | ( ) | ( ) | |
Is the research design appropriate? | ( ) | (x) | ( ) | ( ) | |
Are the methods adequately described? | ( ) | ( ) | (x) | ( ) | |
Are the results clearly presented? | ( ) | ( ) | (x) | ( ) | |
Are the conclusions supported by the results? | (x) | ( ) | ( ) | ( ) | |
Comments and Suggestions for Authors
This is an interesting case report which should be focused on its novel aspect: the bilateral VI nerve palsy
Both the introduction and discussion should be modified to concentrate on this aspect.
Protein S deficiency and puerperium are well known risk factors for CVT, so there is nothing new about puting emphasis on those aspects
Because of that I strongly suggest to modify the tittle and remove "in a pueperium patient with protein S deficiency"
Abstract - line 2 "have presented" should be "present"; lines 8-9 remove the sentence on the prognostic indicators because the statement is not correct. . Multiple venous occlusions and papilloedema are not prognostic indicators for death or disability (see CVT risk score, also available as a free App). Papilloedema is only a por prognostic sign for visual impairment.
Case report - delete all not essential negative or normal findings
Transient numbess were probably focal seizures. CVT do not cause TIAs. Was na EEG performed?
Summarize the follow up
Discussion
remove the mention to TIAs and to the poor prognostic signs
ANSWERS
Title
A part of the title is omitted:... "in a pueperium patient with protein S deficiency"
Abstract
Changes under 1, 2 and 3 are in the sense of a recommended reduction of paper focus on the well-known prothrombotic effect of protein S deficiency:
1. Sentence in row 12 and 13 is left out: "The research has shown the risk period for the CVST is puerperal period".
2. Added to the second sentence of Abstract: „with protein S deficiency“.
3. Sentence in row 15 is left out. „The most significant prothrombotic risk factor is the coagulation disorder caused by the protein S deficiency“
4. According to the reviewer's recommendation, the incorrect statement in lines 19, 20 and 21 was left out: „Despite the prognostic indicators suggested the possible poor outcome (multiple dural venous sinus thrombosis, papilloedema), in this case,“
5. Recommended correction: instead „have presented“ - „present“
Introduction
• Sentence is removed due to recommended reduction of focus on the protein deficiency S: „The prevalence of protein S deficiency in the general European population has been estimated with less than 0.5% [5].“
• Part „caused by protein S deficiency“ of the sentence is removed in sense of recommended reduction of focus on the protein deficiency S.
• According to the reviewer's recommendation that the described symptomatology does not point to TIA but probably seizures, a part of the sentence is left out "similar to the transient ischemic attack-like presentation", and instead of that, another part is placed "consistent with the focal somatosensory seizures"
Case report
• According to a reviewer's instruction for the omission of non-essential findings, the sentence "Lupus anticoagulants and cardiolipin antibodies of IgG and IgM class were negative" was omitted.
• According to a reviewer's instruction for the omission of non-essential findings, part of the sentence is left out: “..... while the protein C activity level was normal at 1.12 (1) (RR 0.74 – 1.49)“.
• According to a reviewer's instruction for the omission of non-essential findings, a sentence is left out: “Thrombocytes, D-dimer and CRP values on the 3rd, 7th and 12th day of the hospitalisation were 580 x 109, 420 x 109 and 278 x 109; 1955, 2110 and 561 μg/L and 21.6, 6 and 1.6 mg/L respectively.”
• Findings of vitamine B12, triglycerides, HDL cholesterol are left out.
• In the text electroencephalograph description is included: “Electroencephalogram during interictal period detected generalized cerebral dysrhythmia without lateralization”.
• Correction: instead of "intracranial pressure" ICP is inserted.
• According to the reviewer's recommendation in terms of reducing follow-up of the course of the disease the sentence is excluded: „A headache was the dominant symptom in the patient, but its intensity gradually decreased after the analgesics were administered“.
Disscusion
• Instead of „caused by protein S deficiency“, „with protein S deficiency“ is placed in the sentence.
• According to the reviewer's recommendation, the TIA discussion was discontinued: „Transient paresthesia and quadrantanopia in our patient were consistent with a transient ischaemic attack-like presentation. Symptomatology corresponded to signs and symptoms related to the anatomical relationship and physiological function of the thrombosed sinus, as explained by O’Rourke et al. [7]. According to a large study on 428 subjects with CVST, stroke-like symptomatology is present in 28.5% [14).
Instead, according to the reviewer's recommendation: „Transient numbness in our patient was consistent with a focal somatosensory seizures.“
• According to the reviewer's recommendation, a discussion of the deficit of protein S was reduced. Therefore, this section was discarded. „Therefore, to the triad described by Makanjoula et al. [15] we could add puerpera. The modified Makanjoula et al. triad would then be: protein S deficiency, pregnancy/puerpera and CVST. The risk of CVST in peripartum period has been reported earlier in several studies, according to which a third [16] or approximately half of female CVST patients were in peripartum period [17].“
• According to the reviewer's recommendation, the sentence is omitted: „In the research conducted by Banakar et al. papilloedema in CVST is a sign of poor outcome in 86% of cases (modified Rankin score (mRS) 3-6) [17]“.
References
Two references from the list are left out:
• Wypasek, E.; Undas, A. Protein C and protein S deficiency – practical diagnostic issues. Adv. Clin. Exp. Med. 2013, 22, 459-467.
2. and 17. (this reference is listed twice in the list by mistake): Banakar, BF.; Hiregoudar, V. Clinical Profile, Outcome, and Prognostic Factors of Cortical Venous Thrombosis in a Tertiary Care Hospital, India. J. Neurosci. Rural. Pract. 2017, 8, 204-208. doi: 10.4103/0976-3147.203812
Extensive editing of English language and style was done.